# Urban Transformation: From Single-Point Solutions to Systems Innovation

**Eleanor Tonks [1,]\* and Sean Lockie [2]**

1   EIT Climate-KIC, 1018 Amsterdam, JA, The Netherlands
2   EIT Climate-KIC, 40129 Bologna, Italy sean.lockie@climate-kic.org
\*   Correspondence: ellie.tonks@climate-kic.org

**Abstract:** Adapting our cities to the new climate regime is critical to ensure that human development is not jeopardized and that the world's citizens can thrive where they live. Faced as we are with the imperative to act, we now need to accept that the challenges we face are not technical in nature—they are systemic. Traditionally, investments in low-carbon city solutions have suffered from being small and disaggregated, with a focus on single-point solutions. To truly enable city transformation at scale, we need to completely rewire our approach to urban innovation and implementation. To face our new reality, EIT Climate-KIC works on catalysing systems change through innovation in areas of human activity that have a critical impact on greenhouse gas emissions—cities, land use, materials, and finance—and to create climate-resilient communities. In this paper, EIT Climate-KIC reflects on its key learnings, as an innovation community, on how to apply innovation in service of urban transformation through the application of nature-based solutions.

**Keywords:** systems change; innovation; nature-based solutions; cities

## 1. Introduction

Adapting our cities to the new climate regime is critical to ensure that human development is not jeopardized and that the world's citizens can thrive where they live. The number of weather- and climate-related loss events has been increasing rapidly since 1980, causing a similar increase in the economic damages from these events. Today, the annual number of events approximate 700 as compared to 200–300 in the 1980s [1]. There is no doubt that climate change is exacerbating the vulnerability of cities. Despite increasing awareness of short-term climate hazards, the medium- and long-term hazards of climate change are currently being under-reported and actioned on by cities [2]. By 2050, eight times as many city dwellers will be exposed to high temperatures (amounting to 1.6 billion citizens), and over 800 million people will be at risk from the impacts of rising seas and storm surges [3]. Cities cannot afford inaction

Global targets for emissions reductions set forth in the Paris Climate Agreement have put low-carbon climate-resilient cities high on the political, financial, and social agenda. They are an essential response to climate change and fundamental to achieving the goals of the Paris Agreement. However, urban transformations into low-carbon societies are not only about saving our climate. Multiple benefits arise from this change, including improved health and well-being, cleaner air, employment creation, an opportunity for community renewal, and positively refreshed state–society relationships.

However, so far, the vast majority of action toward sustainable change in cities has arguably been in words—through the creation of policies, strategies, and targets for the slightly too-distant future, established through too many meetings, workshops, conferences, and talking shops, with Climate Action Plans often missing the detail or the financial backing to attract the investment needed to result in change on the ground. Many cities have made ambitious 2050 targets, but the reality is

that these will come too late. With the acknowledgement that the EU should "concentrate on immediate and urgent" [4] climate policies for 2030, we argue that 2050 is moving too slowly.

Faced as we are with the imperative to act, we now need to accept that the challenges we face are not technical in nature—they are systemic. For too long, investments in low-carbon city solutions have suffered from being small and disaggregated, with a focus on single-point solutions. Understandably, most cities around the world are interested in similar, well-tested, and well-proven projects in climate adaptation and mitigation. However, the cities and projects themselves are regularly too small to justify the involvement of financiers, and often lack the technical knowledge and network to develop and implement their projects at scale.

To truly enable city transformation at scale, we need to completely rewire our approach to urban innovation and implementation, working toward 2030 rather than 2050 and implementing system-orientated approaches in preference to point solutions. According to the IPCC's 1.5 Degree Special Report [5], the world requires rapid and unprecedented transformations not just in energy supply and consumption, but in land-use, urban, infrastructure, and industrial systems, in order to avoid the most perilous effects of global warming. To face this new reality, EIT Climate-KIC works on catalysing systems change through innovation in areas of human activity that have a critical impact on greenhouse gas emissions—cities, land use, materials, and finance—and to create climate-resilient communities.

In the context of systems innovation for urban adaptation, we view nature-based solutions (NBS) as an opportunity to progress toward a low-carbon and more resilient city: one that puts citizens' health and well-being at the centre of urban development and design. NBS are actions which use or are supported by nature and its restorative system processes to address societal challenges, while enhancing citizen well-being and socially inclusive green growth [6]. Furthermore, NBS can play a crucial role in increasing the resilience of urban environments in the new climate regime. Despite the opportunities presented by NBS for citizen health, urban liveability, and climate adaptation and mitigation efforts, the deployment of solutions within our cities remains limited. Integrating long-term considerations into planning requires thinking beyond infrastructure solutions and focusing on systemic changes.

In this paper, we reflect on the key learnings from EIT Climate-KIC, as an innovation community, on how to apply innovation in service of urban transformation. First, we present the origins and initial purpose of EIT Climate-KIC when it was established nine years ago. Then, we discuss four key learnings on the need for a systems approach. Finally, we present three NBS case studies in which we see elements of these learnings being implemented in a real-world context.

## 2. An Innovation Community, Nine Years On

EIT Climate-KIC is a knowledge and innovation community established and funded by the European Institute of Innovation and Technology (EIT) in 2010. Our purpose is to tackle climate change through innovation. We draw our purpose from a context in which climate change is advancing fast and its damaging effects are beginning to take hold. We are Europe's largest public–private partnership with this purpose—a growing pan-European community of diverse organisations united by a commitment to direct the power of creativity and human ingenuity at the climate crisis. We bring together large and small companies, scientific institutions, and universities, city authorities, and other public bodies, start-ups, and students. With nearly 400 formal organisational partners from across 25 countries, we work on innovation to mitigate climate change and to adapt to its unavoidable impacts.

To understand our systems innovation approach, it is first vital to reflect on our nine years of experience in tackling climate change through innovation.

In 2009, the context in which we worked was that research leadership in Europe was not being translated into business growth and job creation, and that such leadership could be orientated to focus on tackling some of Europe's most pressing challenges (of which climate change was one). This research leadership was then targeted toward three key areas:

      I.     Tackling climate change through innovation;

II.　　Developing impactful start-ups and new generations of environmental entrepreneurs;

III.　　Working with universities to educate future decision makers.

We have been reasonably successful in this, as we have achieved the following: supported over 1400 innovative start-ups, attracted plus €900 million to those start-ups, leveraged €3.4 billion climate funding, created over 2000 jobs, launched 367 new products and services, and empowered 17,000 participants through our education activities. Furthermore, the innovations we have supported and nurtured are now starting to support climate action. However, a large question remains, did we tackle climate change at the speed and scale that we need? We acknowledge that the answer is probably not. Europe is not on track to its 2050 targets.

## 3. A Systems Innovation Approach

We understand that most systems we need to transform behave in complex adaptive ways. This means that there are no *right solutions* and that deterministic interventions are bound to fail [7]. Instead, we believe that the best way to shift these systems is through exploration and experimentation, testing ourselves forward through real-world experiences [8]. Now, in 2020, the context in which we work is set out in the IPCC Summary for Policy Makers: "Limiting the risks from global warming of 1.5 °C in the context of sustainable development and poverty eradication implies **system transitions** that can be enabled by an increase of adaptation and mitigation investments, policy instruments, the acceleration of technological innovation and behaviour changes" [5].

Our new strategy (Transformation, in time [9]) is underpinned by four core lessons from our past experiences, which can also be contextualised further through the lens of urban-resilience.

### 3.1. We Cannot Treat Innovation as Techno-Centric

Over the past nine years, we have realised that a supply-driven innovation approach leads to a techno-centric pipeline characterised by incremental improvements of single-point solutions—and rarely to systemic solutions. EIT Climate-KIC is not alone in treating climate change as a complicated problem [7]. Ever since climate change entered the political stage [10] in the 1970s efforts to stem global warming have mostly focused on developing technical solutions through research and engineering. Furthermore, adaptation has often been framed as a purely technical issue that can be addressed through climate-proofing interventions rather than something integral to how city systems function. This means that we continue to see escalating risk and exposure within urban developments, for example, new developments in high risk locations, insurance priced in ways that does not account for positive adaptation actions, or mortgages that are not systematically pricing in future climate risk. This techno-centric approach has produced many important building blocks of a sustainable future (such as renewable-energy technologies or advanced batteries); however, what we must learn now is how to weave these technological advances into our societal systems, along with other cultural, institutional, social, and economic innovations. At EIT Climate-KIC, we define systems innovation as integrated and coordinated interventions in economic, political, technological, and social systems, and along whole value chains.

### 3.2. We Cannot Work on a Project-by-Project Level

Instead we must orchestrate the development and deployment of interventions in portfolios. Portfolios are collections of deliberately chosen innovation experiments. These experiments—representing the supply side of innovation—come in the form of diverse and coordinated innovation projects, education programs, start-ups, ecosystem building activities, citizen engagement strategies, and communication initiatives. They focus on different parts of the same problem, on different opportunities to leverage change. On considering a city's adaptive capacity, we need to shift the locus of interest from individual interventions to the aggregate level, to consider, for example, density and risk exposure, infrastructure vulnerability and resilience, governance, and institutional capacity. One of the distinguishing features of portfolio-based innovation is that, rather than focusing on individual experiments, we seek to understand the aggregate impact of the portfolio.

*3.3. We Need to Demonstrate What Is Possible*

To both inspire and legitimise climate action, we must work with the demand side: governments, corporations, and other challenge owners that share our ambition for transformative action. Ambitious demand-side actors have a strong appetite for change and appreciate the full complexity of the problem, trade-offs or synergies, and the necessary scale for intervention. They are also well-positioned to identify a broad and diverse set of points in the systems they aim to transform, whilst holding the networks to pull through the connected, system-wide solutions we need. EIT Climate-KIC is working at the whole city, region, and supply-chain scale through our Deep Demonstrations programme on systems transformation [11].

*3.4. We Need to Broaden the Innovation Tent*

No single individual, organisation, or government can address the climate crisis alone. Furthermore, the climate risks urban populations face are not only environmental; they raise equity, social justice, and sustainable development issues. To ensure the transition is socially just, diverse communities need to be engaged in decisions made about the future direction of resilience in their towns and cities. For example, involving trade unions, using tools of citizen monitoring and science, and engaging social movements, youth or community organisations, on specific issues, can embed adaptation activities in a socially just resilient transition and contribute to building a politics for transformation. EIT Climate-KIC orchestrates close to 400 innovation partners—from the private, public, and academic sectors—operating as a diverse community that understands the specific needs of different places and contexts. As a community, they bring their own expertise and experience to the systems in which we intervene.

## 4. Discussion

In the case of the built environment, we see nothing but opportunity, but the scale of change is not happening fast enough, nor is it being framed correctly. With political will and the right policies and incentives in place, NBS can be integrated in new developments from the outset (seen through the Sponge City initiatives in China). Retrospectively including them in existing, densely built-up city districts on public and private land and connecting them on the level of the whole city, however, requires working at scale in a long-term mindset (seen in the City of Copenhagen Cloudburst Plan, launched in 2012 and Philadelphia's Green City, Clean Waters programme, launched in 2011). Below, we explore three cities where nature is helping to catalyse change at the scale needed.

The Wild West End Project [12] in London (UK), a collaboration between public authorities, property owners, NGOs, and the private sector, seeks to create a network of green corridors that can sustain biodiversity and improve ecological connectivity. Taking an aggregate view, this diverse consortium is implementing a portfolio of solutions, including green and biodiversity roofs, living walls, sustainable drainage systems, pocket parks, street trees, terrace planters, and pop-up green spaces, to not only address biodiversity, climate, and micro-climate challenges, but also societal well-being. Between 2016 and 2018, the project saw a 17.5% increase in documented multi-functional green infrastructure measures (54 interventions) across the West End. The project builds on a previous Ecology Masterplan, adopted by The Crown Estate, developed to connect two major parks (Regent's Park and St James's Park) in London's West End, by adopting a long-term, estate-wide approach of installing NBS.

Building on the success of the "My tree - My city" public donation campaign [13] (launched as part of the European Green Capital program in 2011), Hamburg has developed a Green Power Strategic Plan to implement a citywide network of green spaces by 2030 [14]. The plan will connect the city's outer ring with its dynamic centre through a series of green axes (walking- and cycling-friendly regenerated or rewilded habitats along parks and rivers). The green network will enhance urban resilience (by providing flood mitigation for vulnerable spaces around Hamburg's vital ports, and by reducing urban heat island effects), improve air quality, and provide numerous health benefits for citizens.

A third project working at scale toward transformation is the Green City, Clean Waters programme [15] (launched 2011) led by the Philadelphia Water Department, on integrating Philadelphia's water systems to safeguard the ecological and economic future of the city. By implementing a range of measures, such as building green storm water infrastructure on public and private land, creating recreational spaces, and restoring biodiversity along water bodies, the programme aims to enhance the region's waterways by managing stormwater runoff and reducing dependence on additional underground grey infrastructure. The city is investing $2.4 billion over 25 years to manage more than one-third of the impervious cover within areas of the city served by combined sewers, a decision supported by a triple bottom-line analysis, comparing the green infrastructure approach with traditional alternatives. Through the Green City, Clean Waters programme, the aim is to generate new employment opportunities; increase $390 million in property value of homes over 45 years; avoid 140 fatalities due to heat reduction over the next 45 years; avoid 250 missed days of work or school per year; and eliminate 1.5 billion lbs of carbon dioxide emissions.

Each of these external case studies can be used to frame, in the context of NBS, the learnings from EIT Climate-KIC in real-world settings. All three view the need for increased application of NBS as not only a climate adaptation measure (a point solution), but rather adopt systems approaches when considering the magnitude of benefits from their interventions (as seen in the indicators of the Green City, Clean Waters programme). Each programme has the ambition to work at scale with a number of different intervention types (as seen by the Wild West End portfolio of solutions), with the support of the demand side (ranging from private land owners to local councils to municipalities to water departments), providing legitimacy and long-term direction for the programmes. Both the Wild West End and "My tree - My city" purposefully seek to engage with a range of stakeholders (from urban planners to local citizens to conservations to children and students). To differing degrees, these cases present practical examples of neighbourhood and city-scale transformation ambitions in the context of urban adaptation. They are not, however, presented as exemplar urban transformations but are instead given as examples through which the learnings from EIT Climate-KIC can be contextualised within an urban context.

## 5. Conclusion

Decision makers working toward urban resilience should consider these lessons learned from EIT Climate-KIC and the external case studies when seeking to address the risks they are exposed to. First, do not approach adaptation as a purely technical issue that can be solved through techno-centric innovation. Second, rather than focusing on the project-by-project or hazard-by-hazard scale, hold space at the aggregate to look across interventions or developments (taking a portfolio approach at scale). Third, to inspire and legitimise climate action work with the demand side and problem-owners that want to take on the mandate for change. Finally, to ensure the transition is socially just diverse communities need to be meaningfully engaged in decisions made about the future direction of resilience in their towns and cities.

**Author Contributions:** Conceptualization, S. L. and E.T.; methodology, S. L. and E.T.; investigation, E.T.; resources, E.T.; writing—original draft preparation, S. L. and E.T.; writing—review and editing, E.T..; project administration, E.T.

**Funding:** This research received no external funding.

**Conflicts of Interest:** The authors declare no conflict of interest.

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
