# Peer review of "Urban Transformation: From Single-Point Solutions to Systems Innovation"

_climate, doi:10.3390/cli8010017_

Round 1

Reviewer 1 Report

The manuscript is presented not as a research work but as an interesting presentation of the opportunities offered by the EIT-Climate Kic program for the application of the research found to industrial development in the field of urban resilience.

The knowledge of this Program may be of considerable interest to a wide audience, both for the methodology of application of the project ideas, and for providing precise indications to the sector's specialists to understand the needs of real applications to civil society.

I therefore suggest its publication precisely in view of the need to proceed with concrete actions concerning urban resilience.

Reviewer 2 Report

I think this kind of knowledge can improve the urban adataption tools research

and it is important the technology transfer between research and companies involved in EIT Climate-KIC.

Reviewer 3 Report

The authors have made a few very minor changes and integrated the idea of Nature Based Solutions, but I still have trouble understanding the purpose of this piece. To summarize, the authors explain that climate change is a major societal problem for which small technical solutions are insufficient. I agree. The authors also propose that EIT Climate-KIC has positioned itself to address these problems as a network of organizations across Europe. Certainly seems like it could be the right approach. Then in section 3 the authors list several theoretical approaches to addressing the systemic challenges of climate change, which seem OK but unoriginal and unsupported by specific examples. Then, the authors list three very short cases of places that have used NBS to address climate change. None of them appear to be associated with EIT Climate-KIC, but rather framed as examples that can help guide EIT Climate-KIC in the future. 

So while I admire the ambitions of EIT Climate-KIC and I have faith that a collaborative, multiorganizational approach to addressing climate change is necessary, the piece still offers no evidence that EIT Climate-KIC has achieved anything pathbreaking. Perhaps the authors wish, instead, to discuss what the organization has learned over the past 9 years? If this were the case, I would hope to read about some specific examples, associated explicitly with EIT Climate-KIC, that show how the authors reached the theoretical lessons: was there an old approach that failed? Were there specific experiments that didn't work?

In summary, I don't think the authors present anything substantially new that isn't already presented on the organization's (very informative) website. 

Reviewer 4 Report

Sorry, I remain unconvinced.

This manuscript is a resubmission of an earlier submission. The following is a list of the peer review reports and author responses from that submission.

Round 1

Reviewer 1 Report

I believe your fundamental thesis is correct: that "investments in low-carbon city solutions have suffered from being small and disaggregated, with a focus on single-point solutions."  For that reason, I was looking throughout the paper for examples of locations where at least a few cities have gone in the opposite direction (as you say, "completely rewire our approach to urban innovation and implementation") to provide greater insight into what is possible (and optimal).

What I found in the Discussion section were three examples, all of them nice and worthwhile, but far from the transformation you are advocating.  They looked like just more relatively small, single-point solutions, rather than the transformation you tout.

Until you can provide more compelling evidence, I'm afraid you are still working in the realm of words, rather than meaningful actions

Author Response

Thanks to the review we have re-emphasised the core thesis, that ‘investments in low-carbon city solutions have suffered from being small and disaggregated, with a focus on single-point solutions…. For this reason a systems approach to urban transformation is needed.’ We have therefore improved the introduction and discussion sections in light of this.

We have also better contextualised, within the four learnings of EIT Climate-KIC, the three external case studies presented in the paper. These are external case studies not funded by EIT Climate-KIC, we agree that there are few real world case studies that address the scale of change needed that we argue for.

Reviewer 2 Report

This short perspective piece describes the systemic challenges of responding to- and adapting to- climate change. To summarize, the authors' argument is that today's (and yesterday's) single-point solutions are poorly matched to the systemic catastrophe that is climate change, and that achieving international goals related to emissions in time to avoid irreversible environmental damage will require 1) a pivot away from old techno-centric approaches, 2) seeing the 'big picture', and 3) demonstrating what is possible, and 4) including more voices in the process.

I don't disagree with this perspective. I think these are generally effective ways at stimulating systemic change.  Unfortunately, I think this piece suffers from major issues. What follows is an honest critique that I hope the authors interpret constructively. Ultimately, I recommend that they withdraw their article and start from scratch. I believe this would be in the interest of the authors, their institution, and this journal.

In summary, the thoughts of this piece are not particularly new, the intention of the authors is unclear, and the language is at some points so abstract that I suspect the piece’s real objective is to obfuscate EIT-Climate-KIC’s lackluster success. It is also obvious that the two authors worked independently—they need to do a better job creating a single narrative voice with a single, clear argument.  I detail these observations below. 

Firstly, I don’t think the ideas in this article are fresh enough to merit a perspective piece. The authors devote a lot of their writing to calling for systemic (rather than piecemeal) change, or "real solutions on the ground that fundamentally changes [sic] the way our cities and societies meet their needs". This is, by now, a stale argument. The pioneers of sociotechnical transitions literature and Transition Management have been arguing for systemic change for two decades (e.g. Rip and Kemp, 1998) and their acolytes have even developed an entire journal for it (Environmental Innovations and Societal Transitions, started in 2012). Their ideas have been applied in the specific realm of cities and urban planning by many many scholars. If the authors of this piece aspire to offer a new perspective, they need to explain why theirs is different or somehow improves upon these well-established perspectives. Perhaps they bring together different actors, use different tools, different frameworks, or achieve better results? In any case, they need to acknowledge that arguments about systemic change are not new, and explicate how they’re pivoting away from them. This may require some critique. Fortunately, I think Transition Management and similar transitions-inspired approaches to change have achieved mediocre results. As the authors correctly observe: real change still awaits. So I believe there are real grounds for something new.  

The authors also seem to be writing a promotional piece or a mission statement for EIT-Climate-KIC, which is mentioned several times in the middle of the article, but invoked very late in the introduction and not at all in the conclusion. If the intention of the authors is to explain how EIT-Climate-KIC is achieving (or planning to achieve) systemic change, the authors need to be clear about this from the beginning (i.e. the first few sentences) and reflect on it in the conclusion. Secondly, if describing the novel approach of EIT-Climate-KIC is the authors’ intention, the authors need to a better job at describing specific examples of how this approach has been applied, and how EIT-Climate-KIC’s framework has contributed to these efforts. The article lists several interesting initiatives in the Discussion (section 4), which I think is the best part of the article. Are these initiatives connected to EIT-Climate-KIC? I’m unsure. I can imagine how the authors might revise the article to explain how their organization’s approach is inspiring these innovative projects, but the connection needs to be explicit and logically air-tight. Lastly on this point, the authors claim that EIT-Climate-KIC “fundamentally…has achieved everything we were supposed to”. Clearly this is not the case—and you acknowledge as such in the next line. I would eliminate this claim.

Finally, there are issues with unsubstantiated assertions throughout the piece, and I fear that this challenge may require the authors to withdraw their piece and resubmit a new piece altogether. They are frankly, irresponsible. For example, in the beginning the authors discuss the urgency of climate change by offering examples of “weather and climate related loss events”. If you use this example, I think you need be specific  about how “weather” events are different from “climate” events, and how we can be sure that the growing number of these events aren’t due to human decisions, e.g. urban growth in disaster-prone areas. I also think the authors can provide more relevant and more recent examples of how climate change threatens cities. The 2011 floods in Copenhagen were over 8 years ago, and might have been unprecedented (if you stick with this example, you need to make the case that this was specifically related to climate change), but they seem trivial compared to heatwave-related deaths around the world, today’s fires in California and Australia, fires in Sweden last summer, and coastal areas that are experiencing inundation as I write this.

On page 1, line 36, the authors make flippant claims about plans and planning efforts that need to be better supported. The critique I interpret is that everyone plans and talks too much, and that real physical changes on the ground are long overdue. I agree with the latter half of this argument, but I also believe the blame is misplaced. Good plans and good planning have never been more important amidst decisions about the built environment that will necessarily be interdependent, irreversible, indivisible, and with imperfect foresight (see work by Lew Hopkins for insight on the importance and role of plans). If anything, planning needs to be enhanced and made more accessible, more democratic, even more grassroots. So while plans written years ago have not necessarily come true, I believe there are other reasons for slow progress than the plans themselves. Furthermore, line 174 references a plan as a success story.

The report drops the word “resilience” a lot, but I don’t recognize that piece’s arguments are related to resilience. As such, its use seems like a hastily placed buzzword.

The use of metaphor throughout the piece is counterproductive. What is “tackling” climate change? This could mean anything, and be interpreted by readers to mean whatever they want.  Line 118 refers to the “fabric of society” and implies that technology has not been weaved into society well enough. This needs to be better substantiated. Renewable energy, battery technology, IOT, and biofuels all seem to be well on their way, but if the “fabric of society” is not absorbing them, I suggest you specific precisely what you mean by this.

Page 3, line 110 seems to indict research, as in all research. This is puzzling to me. Maybe certain types of research are better suited to address systemic problems than others, but research writ-large seems integral.

Around line 122, you discuss portfolio-based innovation rather than project-by-project innovation. You need to provide at least one example of this, and how these things are different. Why is a project necessarily smaller than a portfolio?  

Overall, I would suggest the authors reassess their main objectives, and compose an article that supports those objectives with specific examples. This is probably best achieved by withdrawing their submission and starting from scratch.  

Author Response

Thanks to this review we have critically re-read and amended the paper, improving the content of the introduction, discussion and conclusion.

With regards to the comment on bringing a fresh perspective: The first note to point out is that this is not a new piece of academic literature, rather it is a reflection of the key learnings from an innovation community that has been in place for nine years. We have made this more clear in both the abstract and introduction. We would also argue that though the discussion on systems change is not new there are very few organisations trying to implement this on the scale and ambition that Climate-KIC is bringing to the challenge.

Reference to EIT Climate-KIC and the case studies: as stated above we have now stated quite clearly what the focus of the paper is. Also now stating that the case studies are external, none of these have been funded by EIT Climate-KIC. Our new deep demonstrations programme is the space through which we are implementing systems change but this is too new to provide concrete examples for the readers of the article. We have also re-phrased our assessment of the success of EIT Climate-KIC in line with the comments.

Reference to unsubstantiated assertions: we have amended the text and introduction section when introducing the topic, and showing how climate change threatens cities. 

Reference to planning & line 36: we agree that planning needs to be enhanced and made more accessible, more democratic, more grassroots. However, our text here is not supposed to indicate that people plan to much but rather that changes on the ground must occur at a pace and scale much greater than current rates of change. The intention is that this text will resonate with practitioners that are living these slow, unambitious planning cycles.

Resilience: we have also provided more of a red thread around the focus on nature-based solutions throughout the paper, hence the choice of the focus on resilience. Including a paragraph in the introduction and adding text to the discussion.

Tackling climate change/fabric of society: we have now stated clearly in the abstract, introduction, and in section 2. what the mission of EIT Climate-KIC is. This provides further clarity to readers on what 'tackling climate change' means. We have also reviewed the choice of the phrase 'fabric of society' and removed this from the text.

Indict research: the intention of this statement is not to indict research, rather to reflect on the impact of our innovation portfolio from the past nine years which has been heavily orientated towards research (thanks to the work of EIT Climate-KIC and other climate innovation accelerators which have traditionally viewed innovation in the lens of technology). At the end of this section we emphasise the need to integrating these tech orientated approaches in with other systems (e.g. business models that underpin them, the financing options, social innovation).

Portfolio innovation: Now in the discussion we better link the case studies to the portfolio approach.

Reviewer 3 Report

The present work regards to the urban transformation from an innovation and strategy point of view with respect to the impacts due to human activities. Describes the work of the EIT Climate-KIC, a knowledge and innovation community funded by the European Instite of Innovation and Technology in 2010.
Although it is not a scientific research work in the frame of the special issue "Urban Climate and Adaptation Tools" I think it is an important  issue for the technological transfer between research and local authorities and to give supports to decision makers for implementation decisions in the urban environment.  I therefore believe that in this sense it can be accepted.  

Author Response

Thanks to the review we have re-emphasised the core thesis, that ‘investments in low-carbon city solutions have suffered from being small and disaggregated, with a focus on single-point solutions…. For this reason a systems approach to urban transformation is needed.’ Finally, we agree that this is not a new piece of academic literature, rather it is a reflection of the key learnings from an innovation community that has been in place for nine years, providing key insights for technological transfer between research and local authorities. We have made this more clear in both the abstract and introduction.

Reviewer 4 Report

The manuscript is presented not as a research work but as an interesting presentation of the opportunities offered by the EIT-Climate Kic program for the application of the research found to industrial development in the field of urban resilience.

The knowledge of this Program may be of considerable interest to a wide audience, both for the methodology of application of the project ideas, and for providing precise indications to the sector's specialists to understand the needs of real applications to civil society.

I therefore suggest its publication precisely in view of the need to proceed with concrete actions concerning urban resilience.

Author Response

Thanks to the review we have re-emphasised the core thesis, that ‘investments in low-carbon city solutions have suffered from being small and disaggregated, with a focus on single-point solutions…. For this reason a systems approach to urban transformation is needed.’ We agree that this is not a new piece of research, rather it is a reflection of the key learnings from an innovation community that has been in place for nine years, providing key insights for action between research, local authorities and the needs of civil society. We have made this more clear in both the abstract and introduction.